# In Silico and Chromatographic Methods for Analysis of Biotransformation of Prospective Neuroprotective Pyrrole-Based Hydrazone in Isolated Rat Hepatocytes

**DOI:** 10.3390/molecules29071474

**Published:** 2024-03-26

**Authors:** Alexandrina Mateeva, Magdalena Kondeva-Burdina, Emilio Mateev, Paraskev Nedialkov, Karolina Lyubomirova, Lily Peikova, Maya Georgieva, Alexander Zlatkov

**Affiliations:** 1Department of Pharmaceutical Chemistry, Faculty of Pharmacy, Medical University—Sofia, 2 Dunav Str., 1000 Sofia, Bulgaria; e.mateev@pharmfac.mu-sofia.bg (E.M.); lpeikova@pharmfac.mu-sofia.bg (L.P.); mgeorgieva@pharmfac.mu-sofia.bg (M.G.); azlatkov@pharmfac.mu-sofia.bg (A.Z.); 2Department of Pharmacology, Toxicology and Pharmacotherapy, Faculty of Pharmacy, Medical University—Sofia, 2 Dunav Str., 1000 Sofia, Bulgaria; mkondeva@pharmfac.mu-sofia.bg; 3Department of Pharmacognosy, Faculty of Pharmacy, Medical University—Sofia, 2 Dunav Str., 1000 Sofia, Bulgaria; pnedialkov@pharmfac.mu-sofia.bg; 4Department of Occupational Medicine, Faculty of Public Health, Medical University—Sofia, 8 Bjalo More Str., 1527 Sofia, Bulgaria; k.lybomirova@foz.mu-sofia.bg

**Keywords:** biotransformation, chromatographic methods, in silico approaches, metabolism, pyrrole–hydrazone, synthesis

## Abstract

In the current study, chromatographic and in silico techniques were applied to investigate the biotransformation of ethyl 5-(4-bromophenyl)-1-(2-(2-(2-hydroxybenzylidene) hydrazinyl)-2-oxoethyl)-2-methyl-1*H*-pyrrole-3-carboxylate (**11b**) in hepatocytic media. The initial chromatographic procedure was based on the employment of the conventional octadecyl stationary phase method for estimation of the chemical stability. Subsequently, a novel and rapid chromatographic approach based on a phenyl–hexyl column was developed, aiming to separate the possible metabolites. Both methods were performed on a Dionex 3000 ThermoScientific (ACM 2, Sofia, Bulgaria) device equipped with a diode array detector set up at 272 and 279 nm for analytes detection. An acetonitrile: phosphate buffer of pH 3.5: methanol (17:30:53 *v*/*v*/*v*) was eluted isocratically as a mobile phase with a 1 mL/min flow rate. A preliminary purification from the biological media was achieved by protein precipitation with methanol. A validation procedure was carried out, where the method was found to correspond to all ICH (Q2) and M10 set criteria. Additionally, an in silico-based approach with the online server BioTransformer 3.0 was applied in an attempt to predict the possible metabolites of the title compound **11b**. It was hypothesized that four CYP450 isoforms (1A2, 2C9, 3A4, and 2C8) were involved in the phase I metabolism, resulting in the formation of 12 metabolites. Moreover, docking studies were conducted to evaluate the formation of stable complexes between **11b** and the aforementioned isoforms. The obtained data indicated three metabolites as the most probable products, two of which (**M9_11b** and **M10_11b**) were synthesized by a classical approach for verification. Finally, liquid chromatography with a mass detector was implemented for comprehensive and summarized analysis, and the obtained results revealed that the metabolism of the **11b** proceeds possibly with the formation of glucuronide and glycine conjugate of **M11_11b**.

## 1. Introduction

The hydrazide–hydrazone derivatives are a large group of varied structures. They are characterized by a wide spectrum of biological effects [1]. One of the main pharmacological activities performed by them is the antitubercular effect, which started the path with parent hydrazide Isoniazid (ISH). The molecule is the first line in the therapy of *M. tuberculosis*. A reduction in the toxicity is achieved by various scientists by blocking the NH_2_ group via condensation with carbonyl compounds [2,3]. Another established biological effect related to the hydrazone drugs like Iproniazid, Isocarboxazid, and Nialamide is the antidepressant activity. Their mechanism of action is based on the inhibitory potential of monoamine oxidases, thereby increasing the neurotransmitter levels [4]. The activity of neuroprotection drugs is generally characterized by performance of antioxidative, anti-inflammatory, and MAO inhibitory properties [5]. The first two have a direct effect, while the third one is indirect. Due to the blockage of MAO enzymes, the concentration of hydrogen peroxide levels is reduced, which provides neuroprotection, expressed by safeguarding the central nervous cell against injury. Therefore, these drugs are included in the therapy of neurodegenerative diseases [6].

Metabolic profiling is a tool for the identification and quantification of small molecules (metabolites) in different biological media such as urine, serum, or biological tissue extract [7]. This investigation has provided many opportunities to understand the biotransformation of the evaluated compounds, the metabolic reaction, and the enzymes which catalyzed it. The experimental procedure includes a selection of the biological sample, its purification protocol, and the selection of an adequate analytical technique. 

In recent years, a significant advancement in hyphenated techniques for bioanalysis is available, with a crucial aspect in the field of metabolic investigation. Hyphenated techniques refer to the combination of two or more analytical techniques, such as high-performance liquid chromatography coupled with a diode array spectrometer or mass spectrometry. These hyphenated techniques have revolutionized the field of bioanalysis by providing enhanced selectivity, sensitivity, and speed in the analysis of various biological samples, including pharmaceuticals, metabolites, toxins, and biomarkers [8].

The accurate identification and quantification of analytes in complex biological matrices are essential parts in bioanalysis. This is where hyphenated techniques play a vital role. They allow for the separation of complex mixtures, such as biological samples, into individual components and provide structural information about these components through mass spectrometry analysis [9]. Moreover, these techniques have been widely utilized in the pharmaceutical industry for drug discovery, development, and quality control [10]. Furthermore, hyphenated techniques have also found applications in metabolomics, which focuses on the comprehensive analysis of small molecules involved in metabolic pathways. LC-MS offers high selectivity and sensitivity, allowing for the detection and quantification of a wide range of metabolites in different samples. With these developments, it is now possible to profile metabolites of different classes simultaneously in a semi-automated and non-targeted manner [11]. 

Therefore, this study aimed to design and optimize a new workflow of approaches for detecting CYPs metabolites of a pyrrole-based compound. Initially, three HPLC methods (C18, HILIC, and phenyl–hexyl) were evaluated for the identification of possible metabolites in isolated rat hepatocytes, and the most suitable method was validated. Further, in silico simulations indicated the most probable CYP isoforms participating in the metabolic transformation of the title pyrrole. Thereafter, two metabolites were synthesized and fully elucidated. Finally, the results were validated with liquid chromatography coupled to mass spectrometry (LC–MS). 

## 2. Results and Discussion 

### 2.1. Selection of the Target Hydrazone ***11b***


An essential part of the drug development process is the preliminary assessment of the stability of the lead compound in biological media. Any theoretical and/or experimental conditions allowing evaluation of these properties are a promising tool for rejection of the least drug-like molecule prior to progression into the lead-discovery phase in order to justify a drug discovery effort.

Some of our preliminary experiments with compounds possessing a central pyrrole ring and a hydrazide–hydrazone fragment pointed our attention to their promising effects in neurodegeneration therapy research [12,13,14]. This determined as most scientifically interesting the structure of compound **11b**, presented in Figure 1 and showing the multiple points of biotransformation related to molecules based on an N-pyrrolyl hydrazide–hydrazone moiety.

The process of metabolic conversion of xenobiotics is known as biotransformation. In the living organism, these reactions are catalyzed and controlled by a number of enzymes, with cytochrome P450 considered as the main group related to this processing. These enzymes are most often located within the hepatic endoplasmic reticulum and in extrahepatic tissues [15]. On the other hand, the intensity and the duration of the corresponding pharmacological effect of a biologically active molecule is also determined by the rate and the point of biotransformation [16].

As a result of the enzymatic activity of the many isoforms of CYP 450, a variety of active metabolites may be obtained generating molecules with different pharmacological and/or toxicological properties. Thus, understanding how the compounds are biotransformed is critical to determining the impact of these compounds on human health [15].

Based on this, we applied some in silico approaches as explained later, aiming to mark the most probable chemical changes available in the target molecule of **11b**. The corresponding processes are indicated in Figure 1.

### 2.2. Application of the Developed C18 HPLC Method for Identification of the Possible Metabolites in Isolated Rat Hepatocytes

For evaluating the hepatic biotransformation of the selected hydrazone, a developed and validated RP-HPLC method was implemented. This separation technique was used for the assessment of metabolic changes of ethyl 5-(4-bromophenyl)-1-(3-(2-(2-hydroxybenzylidene)hydrazinyl)-3-oxopropyl)-2-methyl-1H-pyrrole-3-carboxylate (**11b**) [17]. Based on their similar structure and physicochemical properties, the latter method was applied for preliminary estimation of the possible metabolite formation in isolated rat hepatocytes. Initially, the hydrazone was incubated for 2 h with a sample collection every 60 min. Thereafter, sample purification was accomplished with protein precipitation. The separation was achieved on Purospher STAR RP-18 (4.6 × 125 mm, particles size 5 μm) with a mobile phase, which consisted of acetonitrile: phosphate buffer pH 3.5: methanol in ratio 35/35/30 (*v*/*v*/*v*). A 20 μL sample of each time interval was analyzed, and the acquired results at the 60th (Figure 2—black line) and 120th (Figure 2—pink line) minutes were registered.

Importantly, three additional peaks were detected on the chromatograms, which corresponded with the expected biotransformational changes. The analysis of the obtained results revealed that the peak with retention time at 3.403 min is of the parent hydrazide **11**, while the other two with retention times at 4.60 and 11.57 min, respectively, are unknown, and further tests are required. 

### 2.3. Development of HILIC Method 

The detailed literature review indicated that a gel chromatography method would be suitable for the separation and determination of polar compounds such as the expected metabolites [18,19,20,21]. Therefore, the next stage was the development of an analytical technique using a HILIC column, which requires a work program with a large amount of organic solvent [22]. Due to the methanol limitation of the column packing (hydrophobic alkyl chain with a diol group), acetonitrile was preferred. In an attempt to select an optimal mobile phase, the appropriate buffers were prepared: 10 mM of ammonium formate adjusted to pH 3 with formic acid; 10 mM of ammonium formate adjusted to pH 5 with acetic acid; and 10 mM of ammonium acetate adjusted to pH 6.8 with acetic acid. Mixtures of acetonitrile and buffer were created with each buffer separately in various volume ratios. The obtained data demonstrated an unsatisfactory retention of the molecules, and poor separation was achieved, which indicated that the column was unsuitable for the current analyses. This might be attributed to the fact that both the hydrazone **11b** and the tentative metabolites have a high log P, which determined their insolubility in the water layer around the column packing. Therefore, the experiment proceeded with the development and validation of a liquid chromatographic method with a phenyl–hexyl column.

### 2.4. Development of Phenyl–hexyl HPLC Method

The method is reverse-phase, but the main advantage of this column packing is the presence of a hydrophobic hexyl chain, which is associated with a phenyl moiety. This allows the forming of not only the bonds in the traditional C18 columns but also π–π stackings with the aromatic fragment of the target molecules, which ensures additional retention in the stationary phase.

Based on the data from the aforementioned octadecylsilane RP-HPLC method, and the conducted literature search [23,24], the development of the analytical procedure started with the selection of an appropriate mobile phase. Due to the lack of limitations and the good shape of the peaks in the presence of phosphate buffer, phosphate buffer pH 3.5, methanol, and acetonitrile in different ratios were selected as components. The results from the performed preliminary analyses are presented in Table 1. 

The obtained results demonstrated the expected long retention at a high percentage of methanol in the mobile phase. It should be emphasized that by applying the mobile phase ratios from the C18 method, the elution is relatively fast but with a poor peak profile. A slight variation in the ratio of buffer to acetonitrile with a decrease of 2% methanol gave a remarkable reduction in the retention time of nearly 12 min without alteration on the peaks’ symmetry. The analysis of the obtained data concluded that the most suitable mobile phase is in the following volume ratios of the components acetonitrile: phosphate buffer pH 3.5: methanol—17:30:53 (*v*/*v*/*v*).

To enable comparative analysis between the developed chromatographic methods, the flow rate, chromatographic temperature, injection volume, and detection wavelengths were kept as follows: 1 mL/min, 25 °C, 20 μL, 272 and 279 nm.

### 2.5. Method Validation

Validation of the method was conducted according to the ICH Q2 guideline. The procedure started with an investigation of the method’s specificity. Therefore, the mobile phase’s vial was analyzed. The absence of the new peaks with retention time after 2 min proved that the method is specific. The linearity of the method was evaluated in the range of 3.75–30 μM (25–200%) (Figure 3).

The regression analysis was implemented to compute the linear equation and correlation coefficient. Their values y = 0.3395x − 0.2983 and 0.9993, respectively, correspond to the ICH requirements. The sensitivity of the method was estimated through the Limit of Detection (LOD) and Limit of Quantitation (LOQ). Based on the Standard Deviation of the Response and the Slope method, both parameters were calculated, and LOD is 0.57 μM and LOQ is 1.73 μM. The obtained data about the Relative Standard Deviation (RSD = 0.244004) of 6 samples at 100% of the nominal concentration determined the method’s repeatability. The equivalent study was repeated on different days by different analysts, and the RSD (0.690433) proved intermediate precision. The accuracy was assessed by analyzing threefold samples in 3 different concentration levels (50, 100, 150%), and the results are summarized in Table 2.

### 2.6. Application of the Developed and Validated Method for Identification of the Possible Metabolites in Isolated Rat Hepatocytes 

To estimate the method’s suitability for investigation of the tentative metabolites in isolated rat hepatocytes, the analytical procedure was validated according to the M10 guideline for bioanalysis. A blank solution of pretreated isolated rat hepatocytes was analyzed for evaluation of the parameters’ selectivity and specificity (Figure 4). 

The appearance of one peak retained at 3.457 min with area below the permissible 20% of LOQ in the obtained chromatogram proved the method’s specificity. The calibration curve and range were estimated in six concentration levels (3.75–22.5 μM), including LLOQ and ULOQ of hydrazone **11b** in biological media. The recovery of the method was confirmed in 100% of the nominal concentration (15 μM), and the result of 95.3% met the criteria of 15% deviation. The carryover parameter was evaluated through analysis of blank solution immediately after the ULOQ (22.5 μM) sample. The obtained data detected a peak area of 0.63 μM, which corresponded to the requirement and indicated the method as suitable for further investigation.

The developed and validated method was successfully applied to reach the main purpose of the current study. The experimental procedure started with incubation of 200 μL of 1 mM stock solution of **11b** for 2 h. Subsequently, 500 μL (100 μM) of the latter sample was diluted to 1 mL with HPLC-grade methanol and was centrifuged at 14.000 rpm for 15 min. A 400 μL quantity of the limpid supernatant was treated with 1 mL of methanol and was centrifuged again. The obtained supernatant with final concentration of 14.28 μM **11b** was chromatographically analyzed after double filtration through PVDF sterile syringe filters (through 0.47 μm and 0.22 μm). A new peak with a retention time of 6.490 min was observed in the chromatogram (Figure 5—black line), while three peaks appeared using the C18 column. Furthermore, the concentration of the parent molecule (**11b**) decreased to 9.97 μM and was retained at 13,087 min. After 120 min of incubation (Figure 5—pink line), two additional peaks (t_R_ = 4.183 and t_R_ = 4.447) were obtained, which correspond to the data from octadecylsilyl analysis.

It should be emphasized that a metabolic reaction had occurred considering the appearance of the new peaks and the increase in the area of peak 4 (t_R_ = 6.480), while the concentration of the selected hydrazone decreased to 7.58 μM after 2 h of incubation. 

### 2.7. Prediction of the Possible Metabolites’ Structures via BioTranformation 3.0 

Over the past decade, in silico approaches have been widely used in drug discovery and the development of leader structures through drug design [25,26]. Nowadays, computational techniques are increasingly included in preclinical studies for the determination of pharmacokinetic parameters (ADME), as well as for the evaluation of toxicity of the parent structure and its impurities [27] and possible metabolite formation [28]. 

Hence, in the current study, the free available software BioTransformer 3.0 was applied to predict the tentative metabolites’ structures, their isotope mass, the metabolic reactions, and the CYP isoforms which catalyzed them. The SDF file of the parent hydrazide–hydrazone **11b** was used, and the prediction was accomplished by the function “Metabolic prediction”, which includes conformation in human CYP enzymes. The obtained results demonstrated the formation of 12 possible metabolites, 10 of which are hydroxylated derivatives with 499.07 isotopic mass, while 2 of them are products of O-dealkylation. Four main isoforms are responsible for these structural changes. CYP 1A2 catalyzes eight reactions, resulting in aromatic or aliphatic hydroxylated products. The aromatic changes in both the 4-bromophenyl core and the side 2-hydroxybenzene were predicted. Aliphatic hydroxylation is possible at the methyl group on the 2nd position in the pyrrole ring or at the ethyl group from the ester moiety at the 3rd position of the poly-substituted pyrrole core. CYP 2C9 and 3A4 also catalyze the aromatic hydroxylation on the 4th and 5th positions of the 2-hydroxybenezen moiety from the carbonyl fragment, respectively. The other possible metabolic pathway of O-dealkylation generates two products identified as 5-(4-bromophenyl)-1-(2-(2-(2-hydroxybenzylidene)hydrazineyl)-2-oxoethyl)-2-methyl-1H-pyrrole-3-carboxylic acid (**M11_11b**) and ethanol (**M12_11b**). Their formation is catalyzed by CYP 2C8. The detailed data are presented in Appendix A.

### 2.8. Application of the Docking Approach

Various isoforms of the CYP450 family are deposited in the Protein Data Bank (PDB) and are primarily used in the molecular docking analysis, which could provide rapid preliminary evaluation of the inhibition or activation of CYP450 isoforms. For the current work, **2HI4** (CYP1A2), **2VN0** (CYP2C8), **5W0C** (CYP2C9), and **2V0M** (CYP3A4) with co-crystallized *alpha*-naphthoflavone, Troglitazone, **9W6**, and Ketoconazole, respectively, were retrieved from the Protein Data Bank due to their good resolutions, as well as standard co-crystallized ligands.

The results obtained from the aforementioned BioTransformer server demonstrated four possible enzymes involved in the metabolism of **11b**. Thus, molecular docking simulations in the active sites of the four CYP450 isoforms of interest (1A2, 2C8, 2C9, and 3A4) were carried out with the docking software Glide (Schrödinger), considering the reliable results of the program in similar studies [29].

However, a diverse set of conformations of the active sites of the used CYP isoforms has been previously described [30], which implies the introduction of simulations with flexible active amino acids. Recent work discussed an 89% success rate after the implementation of an induced-fit docking (IFD) against various CYPs [31]. Moreover, the robustness of the obtained in silico results could be enhanced by MM/GBSA recalculations [32]. Thus, the application of more precise and hardware-demanding simulations, such as IFD and MM/GBSA, was incorporated in the current work. The docking scores are presented in Table 3.

The obtained results demonstrated that the pyrrole-based compound **11b** could fit in the substrate clefts of three CYP450 isoforms—2C8, 2C9, and 3A4. The docking score of the title ligand in 2C8 (PDB: **2VN0**) was close to the docking value of the co-crystallized and active CYP2C8 inhibitor Troglitazone. However, the MM/GBSA recalculations displayed better binding of the native inhibitor compared to **11b**. The pyrrole compound formed moderately stable complexes with isoforms 2C9 (PDB: **5W0C**) and 3A4 (PDB: **2V0M**), but the obtained values after the IFD and MM/GBSA simulations were drastically lower compared to the co-crystallized ligands (**9W6** and Ketoconazole). Interestingly, the compound of interest, **11b**, could not return favorable intermolecular interactions with the active amino residues in CYP1A2. 

Further observations of the distances between **11b** and the aforementioned investigated CYPs were conducted. The obtained poses are considered to provide a possible catalytically active conformation if the ligand’s pose is within 6 Å from the heme iron. We noted that **11b** was situated at 4.67 Å from the iron group of the hem in the active site of CYP2C8 (Figure 6a). The 2-hydroxyphenyl moiety was facing the hem cofactor, increasing the probability of oxidation of the latter.

The distance between **11b** and CYP2C9 was slightly higher (7.65 Å) compared to the optimal value; however, a water-mediated halogen bond was formed in close vicinity to the hem group (Figure 6b). The active conformation in CYP3A4 was located close to the hem group (3.66 Å) with the *p*-bromophenyl moiety facing the cofactor (Figure 6c). Overall, it could be hypothesized that **11b** is metabolized mainly by three isoforms: 2C8, 2C9, and 3A4.

### 2.9. Conventional Synthesis of the Tentative Metabolites and Their Full Chemical Characterization

The comprehensive in silico assessment authenticated the reactions of hydroxylation and O-dealkylation, catalyzed by CYP 2C9, 3A4, and 2C8, as metabolic pathways from phase I. Based on this presumption, two novel hydroxylated compounds were synthesized via conventional condensation between 2 mmol of the parent hydrazide (**11**) and 2 mmol concentration of the corresponding carbonyl partners—2,4 dihydroxybenzaldehyde or 2,5 dihydroxybenzaldehyde. The synthesis of the hydrazide of N-pyrrolyl carboxylic acid (**11**) was accomplished according to a recently reported procedure by Bijev et al. [33]. The reactions were carried out under conventional heating for 30 min using 3 mL of glacial acetic acid as a solvent (Figure 2). When cold water was added, the two newly obtained hydrazide–hydrazones precipitated, and a following recrystallization was conducted in ethanol.

The newly synthesized structures were fully characterized with chromatographic procedures (TLC, HPLC), spectroscopic techniques (IR, ^1^H–NMR), and a combined method (LS–MS). The corresponding structural characteristics are presented in the Materials and Methods section as follows. Their MS2 spectrum are presented in Appendix A.

The synthesized new molecules were subjected to chromatographic evaluation, attempting to identify the characteristic retention times of the target products. The analysis was initially performed on a phenyl–hexyl stationary phase method, applying the developed and validated conditions. The corresponding chromatograms are presented in Figure 7.

The obtained results from the phenyl–hexyl method demonstrated the retention for **M9_11b** at 7.25 min and 7.327 min for **M10_11b**, which did not correspond to the obtained retention times of the probable biotransformational products of the processed compound **11b** under the discussed conditions, as presented in Figure 5, namely t_R_ of 4.2 and t_R_ of 6.5 (Figure 5). This led to the application of an additional LC-MS method for verification of the obtained result.

### 2.10. Application of LC-MS Method for Verification of the Obtained Results 

The hyphenated technique LC-MS is increasingly applied as a component of the evolving “metabolomics toolbox”, with the advantages of speed and largest metabolite coverage with the highest sensitivity [34]. In an attempt to confirm and verify the obtained results from previous investigations, the LC-MS method was employed. The final step of the current research was to analyze samples of the incubated **11b** in isolated rat hepatocytes with an additional comparison of the resulting *m*/*z* and *m*/*z* of the newly synthesized compounds **M9_11b** and **M10_11b** (Table 4).

The obtained mass spectra for the three compounds (**11b**, **M9_11b**, **M10_11b**) demonstrated similar fragmentation. They provided one molecular ion [M + H]^+^ with *m*/*z* 484.0863 for **11b** and 500.0815 for both metabolites. The fragment ions at *m*/*z* 438.0439[M + H–46]^+^, *m*/*z* 438.0439[M + H–46]^+^ indicated the loss of CH_3_CH_2_OH. The unstable R-NH-N=R bond generated the product ions of hydroxy-benzylcation at *m*/*z* 107.0495 Da in the structures of the tentative metabolites.

The comparison of the aforementioned spectral data (Figure 8) with the chromatogram of incubated **11b** revealed the absence of the putative metabolites with their exact retention times and *m*/*z*. This may be because both structures are part of phase I metabolism and may have biotransformed into more polar conjugates. Thus, the server BioTransformer was again applied for the prediction of possible metabolites from phase II metabolism of the compounds **M9_11b**, **M10_11b**, and **M11_11b**. The results demonstrated the formation of 7 metabolites through reactions of glucuronidation and amino acid conjugation catalyzed by glucuronosyltransferase and glycine-N-acetyltransferase, respectively. The detailed data are presented in Appendix A. 

The performed LC-MS evaluation did not show any conclusive results on the structure of the second-phase metabolic products of the synthesized **M9_11b** and **M10_11b**. The obtained m/z values turned our attention towards possible glucuronidation and/or conjugation with glycine of the unidentified **M11_11b**, which is a subject of additional evaluation. The glucuronidation of this metabolite may occur at two positions due to the presence of two functional groups—the OH-group at the second position in the lateral benzene ring and a free carboxyl group in the pyrrole. The obtained quasi-molecular ion [M + H]^+^
*m*/*z* 632.1446 with relative intensity of 16% corresponds to the data from the web server, and the registered base peak with *m*/*z* 631.1419 (100%) corresponds to the molecular mass of the glucuronide. In the same sample was detected protonated molecular ion [M + H]^+^ at *m*/*z* 513.4200, which corresponds to glycine conjugate. A product base peak at *m*/*z* 89.0603 was observed on the spectrum, which may be attributed to the HOOC-CH_2_-NH-CH_3_ fragment.

Additional experiments are planned in order to confirm the observed data.

## 3. Materials and Methods

### 3.1. Formatting of Mathematical Components

All solvents for chromatographic analyses were purchased from Fisher Scientific (Seoul, Republic of Korea) and were of HPLC/MS grade. Buffer’s salts and acids: ammonium formate, ammonium acetate, potassium dihydrogen phosphate, disodium hydrogen phosphate, glac. acetic acid, and orthophosphoric acid were obtained from ThermoFisher Scientific, Milan, Italy. Water used throughout the analyses is from a Millipore system. The mobile phase buffers were prepared according to the European Pharmacopoeia and filtered through a membrane filter. The carbonyl compounds for synthesis, 2,4-dihydroxybenzaldehyde and 2,5-dihydroxybenzaldehyde, were purchased from Acros Organics, Geel, Belgium. The parent hydrazide **11** and its selected hydrazone **11b** were previously synthesized and described in Bijev et al. [33]. 

### 3.2. Chromatographic Parameters 

All chromatographic methods were carried out by UltiMateDionex 3000 SD. For preliminary evaluation of possible metabolites, the chromatograph was equipped with a Dionex UltiMate DAD 3000 detector (ThermoFisher Scientific, Milan, Italy). The latter parameters are summarized in Table 5.

The LS-MS method was performed on the aforementioned chromatographic system associated with a (HESI-II) Q Exactive Plus mass spectrometer (ThermoFisher Scientific^®^). For separating the tentative metabolites, the octadecylsilane column was used with the mobile phase, which consisted of 0.1% formic acid in dist. water (solvent A) and 0.1% formic acid in acetonitrile (solvent B). The analysis was conducted by gradient elution with a constant 0.3 mL/min flow rate of 0–1.5 min, 5–10% B; 1.5–13 min solvent B linearly increased to 60% followed by 6 min isocratic elution; 19–23 min 60–95% B and finished with 95% solvent B an isocratic elution for 2 min for 25 min. 

### 3.3. Preparation of the Working and Sample Solutions 

The stock solution of **11b** was prepared by accurately weighing 2.4 mg and dissolving in a volumetric flask of 5 mL with dimethyl sulfoxide (DMSO) (1 mM). Working solutions for validation were carried out by dilution method (37.5–300 μL) to a final volume of 10 mL with the mobile phase. The sample solutions were obtained after incubation of 200 μL of the stock solution for 2 h, with a final concentration of 100 μM. At the point of taking, to 500 μL of the evaluated media 500 μL of methanol was added, and afterwards, the obtained mixture was centrifuged for 15 min at 14.000 rpm. The formed supernatant was diluted to 14.28 μM with methanol, and the obtained solution was centrifuged again. The clear supernatant was filtered through PVDF sterile syringe filters (through 0.47 μm and 0.22 μm) and subsequently analyzed. 

### 3.4. Isolated Rat Hepatocytes

The investigation was conducted using 200 g male Wistar rats, housed in plexiglass cages at room temperature and acquired from the National Breeding Center, Sofia, Bulgaria. All procedures were approved by the Institutional Animal Care Committee in accordance with European Union Guidelines for animal experimentation. The rat was anesthetized with sodium pentobarbital (0.2 mL/ 100 g), and a modified method described by Fau et al. [35] was applied for the in situ isolation of the cells and the liver perfusion. Next, 100 mL of HEPES buffer (pH  =  7.85) was utilized for the perfusion of the liver. Subsequently, 200 mL of HEPES buffer (pH  =  7.85), and a final addition of 200 mL of HEPES buffer containing collagenase type IV (50 mg/200 mL) and 7 mM CaCl_2_ (pH  =  7.85), were added. A dispersion of the hepatocytes in 60 mL of Krebs–Ringer-bicarbonate (KRB) buffer (pH  =  7.35) was completed after providing a cell suspension of small pieces of the liver. The hepatocytes were centrifuged at 500 g for 1 min after the initial filtration, and they were washed three times with a KRB buffer. Cells were counted under the microscope, and the viability was assessed by Trypan blue exclusion (0.05%), which was calculated to be 89% on average [35]. Cells were diluted with a KRB buffer to make a suspension of about 3  ×  106 hepatocytes/mL. Incubations were carried out in 25 mL Erlenmeyer flasks with each containing 3 mL of the cell suspension (i.e., 9  ×  106 hepatocytes). The latter procedures were performed in a 5% CO_2_  +  95% O_2_ atmosphere.

### 3.5. In Silico Approaches 

#### 3.5.1. BioTransformer 3.0

The free available web server BioTransformer 3.0 was applied for the prediction of tentative metabolites. In the current study, the combined CYP 450 mode was used, which is the default setting on the server and allows for the most comprehensive generation of potential metabolite structures. The SDF format of **11b** was implemented, and after calculation, the data on the metabolites’ structures, their isotope mass, the metabolic reactions, and the CYP isoforms which catalyzed them were obtained. 

#### 3.5.2. Molecular Docking 

Selection and Preparation of Proteins

The crystallographic structures of CYP1A2 (PDB ID: 2HI4) [36], CYP2C8 (PDB ID: 2VN0) [37], CYP2D6 (PDB ID: 4WNU) [38], and CYP3A4 (PDB ID: 2V0M) [39], resolved with the co-crystallized ligands alpha-Naphthoquinone, Troglitaone, Quinidine, and Ketoconazole, respectively, were retrieved from the Protein Data Bank (PDB). The Protein Preparation Wizard in Maestro (Schrödinger Release 2022-3: Protein Preparation Wizard; Epik, Schrödinger, LLC, New York, NY, USA, 2022) was employed for the protein refinement processes. Hydrogen bonds and het states at pH 7.0 were generated followed by the removal of waters not situated in the active site. The energies of the crystallographic structures were minimized by applying the OPLS4 force field.

Ligands preparation

The chemical structure of the applied pyrrole-based compound **11b** was drawn with the 2D sketcher module in Maestro and converted to the corresponding 3D structure with the LigPrep module (Schrödinger Release 2022-3: LigPrep, Schrödinger, LLC, New York, NY, USA, 2022). Utilizing the module, hydrogen bonds, tautomers, enantiomers, and ionization states at pH 7.0 were generated. Furthermore, the ligand’s energy was minimized by applying the OPLS2005 force field.

Docking protocol

The docking module in Maestro–Glide was implemented for the current study considering the reported reliable results when CYP enzymes were simulated with the software (Schrödinger Release 2022-3: LigPrep, Schrödinger, LLC, New York, NY, USA, 2022) [40]. Glide uses the empirically based GlideScore scoring algorithm, which has three options: High-throughput Screening (HTS), Standard-Precision (SP), and Extra-Precision (XP) modes. In the current study, the most precise XP docking mode was applied. Furthermore, the induced-fit docking (IFD) in Schrödinger was implemented to further validate the examined conformations. The IFD examines the active side chains as fully flexible, which leads to optimization of the active site conformation. MM/GBSA (Molecular Mechanics–Generalized Born Surface Area) re-calculations were also introduced to determine the binding free energies of the enzyme–ligand complexes. The grid boxes were generated around the co-crystallized ligands with the Receptor Grid Generation module in Maestro.

### 3.6. Synthesis of the Suggested Metabolites

Synthesis was conducted via conventional condensation between previously obtained hydrazide **11** and a carbonyl compound. The reaction was carried out with equimolar quantities (2 mmol) of hydrazide of the N-pyrrolyl carboxylic acid and 2,4-dihydroxybezaldehyde or 2,5-dihydroxybezaldehyde in 3 mL glac. acetic acid media for 20 min under conventional heating. The precipitation of the obtained hydrazones, **M11b_9** and **M11b_10**, was formed by adding cold water. Recrystallization from ethanol was conducted as a final step. 

Ethyl 5-(4-bromophenyl)-1-(2-(2-(2-hydroxybenzylidene)hydrazinyl)-2-oxoethyl)-2-methyl -1H-pyrrole-3-carboxylate (**11b**): IR (in KBr): 3390 (OH), 3140 (NH), 2840–2950 (CH3 and CH_2_), 1670 (COOC_2_H_5_), 1630 (Amide I), 1560 (AmideII), 1230 (C-O), 830 (p-disubstituted C_6_H_4_), 750 (o-disubstituted C_6_H_4_); 1H NMR (at 600 MHz, in DMSO): 1.27 (t, 3H, CH_2_CH_3_), 2.45, 2.48 [d, 3H, CH_3_(2) *J* = 9.54], 4.18–4.22 [q, 2H, CH_2_CH_3_, *J* = 7.04, *J* = 7.04], 4.68 (s, 1H, CH_2_N), 5.07 (s, 1H, CH_2_N), 6.49 [s, 1H, H(4)], 6.81 [t, 1H, H(3”)], 6.90–6.93 [m, 1H, H(5”)], 7.22–7.H(2’), H(6’)], 7.29–7.31 [m, 1H, H(4”)], 7.56–7.59 [m, 3H, H(6”), H(3′), H(5’)], 8.32 (s, 1H, CH=N), 10.1 (s, 1H, OH), 11.68 (s, 1H, CONH).

Ethyl-(E)-5-(4-bromophenyl)-1-(2-(2-(2,4-dihydroxybenzylidene)hydrazineyl)-2-oxoethyl) -2-methyl-1H-pyrrole-3-carboxylate (**M9_11b**): Yield 95%; Melting point 249.1–249.5 °C; Rf 0.69/(CHCl_3_:C_2_H_5_OH = 10:0.5); IR vmax: 3185 (NH), 2840–2940 (CH_3_ and CH_2_), 1710 (COOC_2_H_5_), 1260 (C-O), 772 (o,p-disubstituted phenyl ring); ^1^H NMR (DMSO, 400 MHz) δ 11.85 (1H, s, OH-2’), 11.60 (1H, s, CO-NH), 10.00 (1H, s, OH-4’), 9.91 (1H, s, CH=N), 7.63 (2H, t, *J* = 8.80, H-2, H-6), 7.30 (d, *J* = 5.80, H-3’, H-6’), 7.25 (2H, t, J = 7.80 Hz, H-3, H-5), 6.50 (1H, d, *J* = 5.10, H-5’), 5.05 (1H, s, H-4 (pyrrole)), 4.20 (2H, q, CH_2_-CH_3_), 3.35 (3H, s, CH_3_), 2.50 (2H, t, CH_2_-C=O), 1.28 (3H, t, *J* = 8.20 Hz, CH_2_-CH_3_); *m*/*z* (FTMS + pESI) 500.08 [M + H+]; HPLC (t_R_) 7.25.

Ethyl (E)-5-(4-bromophenyl)-1-(2-(2-(2,5-dihydroxybenzylidene)hydrazineyl)-2-oxoethyl) -2-methyl-1H-pyrrole-3-carboxylate (**M10_11b**): Yield 95%; Melting point 243.0–243.5 °C; Rf 0.73/(CHCl_3_:C_2_H_5_OH = 10:0.5); IR vmax: 3390 (OH), 3189 (NH), 1663 (COOC_2_H_5_), 1629 (Amide I), 1242 (C-O), 809 (m-disubstituted phenyl ring); ^1^H NMR (DMSO, 400 MHz) δ 11.47 (1H, s, OH-2’), 11.05 (1H, s, CO-NH), 9.98 (1H, s, CH=N), 9.90 (1H, s, OH-5’), 8.25 (2H, t, *J* = 8.10, H-2, H-6), 7.62 (2H, t, *J* = 7.70 Hz, H-3, H-5), 6.50 (1H, s, H-6’), 6.30 (1H, m, H-4 (pyrrole); 1H, H-3’, H-4’), 4.65 (2H, t, CH_2_-C=O), 4.29 (2H, q, CH_2_-CH_3_), 3.50 (3H, s, CH_3_), 1.28 (3H, t, *J* = 8.20 Hz, CH_2_-CH_3_); *m*/*z* (FTMS + pESI) 500.08 [M + H+]; HPLC (t_R_) 7.327.

## 4. Conclusions

In the current study, the prospective neuroprotective compound **11b** was selected for metabolic investigation. To achieve the aim, hyphenated techniques were successfully applied. A novel, rapid RP-HPLC method was developed and validated. The obtained data demonstrated the appearance of three new peaks after 2 h of incubation of **11b** in isolated rat hepatocytes. For comprehensive evaluation, in silico approaches were used for structural prediction of the most possible metabolites, **M9_11b** and **M10_11b**, which were synthesized via conventional condensation. The final LC-MS analysis of the biotransformation of **11b** revealed the absence of the putative structure in a biological sample. In the LC-MS spectra of the incubated hydrazone were observed peaks, which proved that the metabolism of the latter proceeds with the formation of glucuronide and glycine conjugate of **M11_11b.**

The developed multistep strategy can be used in further bioanalyses for metabolite profiling. 

## Data Availability

The data presented in this study are available upon request from the corresponding author.

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
