# Peer review of "In Silico and Chromatographic Methods for Analysis of Biotransformation of Prospective Neuroprotective Pyrrole-Based Hydrazone in Isolated Rat Hepatocytes"

_molecules, 2024, doi:10.3390/molecules29071474_

Round 1

Reviewer 1 Report

Comments and Suggestions for Authors

Future application of this novel method is not clear and needs to be elaborated.  Figures should be of high resolution. For example, the figure 5 the bond angle, font is not clear.

Reviewer 2 Report

Comments and Suggestions for Authors

This paper contained an account of the synthesis of a series of molecules through a one-step condensation procedure followed by their incubation with hepatocytes and subsequent HPLC separation of the metabolites. The components were tested for binding to proteins including by in silico methods to help understand why certain products were formed.

As my own area of expertise is organic chemistry, I can only comment on this with authority, and with less certainty on the other areas (although I am very familiar with chromatographic purification methods).

Although I can see that there is a good deal of work in this paper, I do have a significant problem in understanding the significance and context of the results, primarily because of the way that the paper is presented.

First, a compound ‘11b’ was prepared and this was then incubated with hepatocytes, with samples taken and analysed by a HPLC method. However, there is no diagram of any molecules (the author should not have to work out what it is from the name alone) until Scheme 1 on page 9 of the manuscript. However, the diagram does not contain a molecule labelled ‘11b’ and indeed none of the illustrated molecules are numbered – is this one of the products illustrated? The text states that the small organic molecules were characterised by 1H  NMR, IR and LS-MS. However, I could not find this data in the paper or supporting information (other that some HRMS data). The supporting information lists multiple similar molecules which appear to have the same molecular mass in most cases but a different substitution pattern – but how were the structures of these proven? I cannot find spectroscopic data to support the proposed structures.

I would recommend that the authors significantly revise their manuscript by first stating clearly what molecules were selected for incubation, with a clear illustration of the molecule, how it was prepared and characterisation data to confirm its structure. Then the incubation method should be clearly explained with details of how the products were extracted, and their suspected structures illustrated in the main paper, not just the SI. The SI should contain the characterisation data for each suspected metabolite, to confirm their structures. The HPLC methods are described in a good level of detail, although they could be clearer. For example, on page 6 there are two chromatograms which are very blurred and difficult to read, even on expansion. What is the very large peak at ca 13 minutes?

In a typical paper containing small molecules, numbering tends to start at ‘1’ and increases. However, in this paper there just seems to be ‘11b’ and nothing before it. Where are compounds 1-10? Furthermore, the compound numbers should be in bold font.

So, at this point, and the current form of the paper, I can’t really assess what was carried out and what the products were. I would recommend that the authors submit a revised paper addressing the points above and I would be happy to review it.

Comments on the Quality of English Language

Generally good but could benefit from a careful check by an expert proof reader.

Reviewer 3 Report

Comments and Suggestions for Authors

The manuscript of Mateeva et al. presents a study on the metabolism of an hydrazone derivative 11b using untreated rat hepatocytes.

In 2021, I expect a metabolic study to present the essential metabolites based on at least mass data.  In this manuscript the authors have first validated a good UV HPLC chromatographic method able to show disappearance of the substrate 11b and appearance of some metabolites for which they do not show the UV spectra and don’t know the structure.  But that method, using phosphate, is not compatible with HPLC/MS.

Then in a later part they have studied the metabolism using an HRMS HPLC/MS system (a Q-exactive) but completely different solvents (compatible with ESI MS). I don’t know how much time they have spend for this part.

But it looks that they got preliminary interesting results but that are hard to relate to the first part.

They give a few mass values and MS2 values obtained on this system. I would have appreciated to see the chromatogram (UV and TIC) and reconstituted chromatogram for each of the m/z value detected. Then a MS2 spectrum for each metabolite could have been shown. The value given for m/z of the parent and 2 hydroxylated metabolites are perfect. The value given for the glucuronide (could be the acyl glucuronide) however are problematic since the m/z is at least 30 mmu different from the calculated value. Same for the m/z of the glycine conjugate that is also in error of about 40 mmu.

This should have been better checked.  I suggest that the authors show the reconstituted chromatogram for each specific m/z and show the mass spectrum and the MS2 spectrum somewhere. (in the paper or in the supplementary)

The authors have synthetized two potential hydroxylated metabolite (on the phenyl group of the benzylidene ring. They should have shown if these authentic compounds co-elute in the UV HPLC with some metabolites  and also for the HPLC/MS and they should give their HPLC/MS data.. 

Also about MS your compound 11b is brominated so it show a doublet in MS at M+H+ and M+H+ plus 2. This can be used to extract all the metabolites that contain bromine. There is a way of analyzing the raw data for isotopic ratio of m/z and m/z+2. Check with your mass specialist. Using isotopic filtering decreases very much the background and allow you to see very minor metabolites. (This is particularly doable with an HRMS instrument like the Orbitrap.) Evidently glutathione conjugates replacing the bromine would not be seen. They could be perhaps found using neutral loss of 129.

The Docking part:

The authors have used a good software to dock the parent compound 11b in some X-ray structure of several human cytochrome P450. They have choosen some PDB structure without explaining why and not citing the literature of the used templates. All these structures in the PDB have been obtained by scientists through had work and their papers deserve to be cited.

Now if you do not know the metabolites of 11b docking will not tell you the productive positions of which enzyme. Moreover you have used rat hepatocytes and thes study docking in human enzymes that are analogous but different from rat enzymes. So that your data cannot be extrapolated to human without checking.

Along the paper I have done many remarks that I will enumerate below.

            Page 1 : It would be important to give a scheme of the metabolism in the paper not only in the supplementary.

What is the structure of the starting drug 11b? Show it.

            Page 2 Results: the paper shows a good HPLC UV method using phosphate buffer (methods used before 2005). Now most scientists use HPLC(UV)MS methods at least for identification of possible metabolites. You should also give more data on your HPLC/MS system. For HPLC/MS phosphate is proscribed. One generally use ammonium acetate (or formiate) buffer. (you used formic acid)

            Page 4 table 1 : Why do you use isochratic conditions.  In general when analyzing metabolites one use a gradient so that one can see the polar metabolites and also the less polar ones on the same chromatogram. Glucuronide, sulfates, the hydroxylated metabolites, carboxylic acids, esters and so on.

            Line 178 : Figure 3 : Blank solution

            Page 6 line 189 and further: What is the concentration of the incubation. You do not give the final volume? When you describe the hepatocyte incubation you should give the total volume (I believe 3mL in a 25 ml well) and the final concentration of substrate.

The best would to show some kinetics. Here you show 60 and 120 min but don’t really tell what is increasing with time. Could you superimpose the chromatogram (with a different color or pattern) of fig 1 and simply write the time on each line. You could also probably put the zero time on the same figure. Thus some kinetic would be evident. Then name the peaks (S for substrate then M1;, M2 , M3 … for metabolites. This would be easier for the discussion. In the same system what are the retention time of your synthetic hydroxylated compounds? Is there a variation in the UV spectrum?

            Line 250  Docking: PDB models generally correspond to a literature reference. thus 2VNO corresponds to: Determinants of cytochrome P450 2C8 substrate binding: structures of complexes with montelukast, troglitazone, felodipine, and 9-cis-retinoic acid. Schoch, G.A., Yano, J.K., Sansen, S., Dansette, P.M., Stout, C.D., Johnson, E.F.

(2008) J Biol Chem 283: 17227-17237 that should be cited 

Same for 5WOC : Determinants of the Inhibition of DprE1 and CYP2C9 by Antitubercular Thiophenes.Liu, R., Lyu, X., Batt, S.M., Hsu, M.H., Harbut, M.B., Vilcheze, C., Cheng, B., Ajayi, K., Yang, B., Yang, Y., Guo, H., Lin, C., Gan, F., Wang, C., Franzblau, S.G., Jacobs, W.R., Besra, G.S., Johnson, E.F., Petrassi, M., Chatterjee, A.K., Futterer, K., Wang, F. (2017) Angew Chem Int Ed Engl 56: 13011-130152V0M : Structural Basis for Ligand Promiscuity in Cytochrome P450 3A4 Ekroos, M., Sjogren, T.

(2006) Proc Natl Acad Sci U S A 103: 13682. Your choice for the stated structure is not evident since there are many structures od CYP3A4 and that of Ekros is on a mutant, I think, on a loop between helix G and F.

            Figure 5 : Is it so important to do modelisation of docking in active sites when you have not identified the CYP that metabolizes your substrate? And you are not certain of all metabolites. Classically one should start doing correlations using different human microsomes or, and use inhibitors to try to find which human P450 forms which metabolite. Then it is legitimate to do docking studies.

Here you have learned how to use a good software. This is didactic. But the important thing if your  11B becomes a drug is to know its bio-disposition and its metabolites.

            Table 4 and below 486 : This is the other isotope 18Br; 438 corresponds to loss of ethanol giving the ketene.  107 may correspond (using Chemdraw)  to   C7H7O+ (Exact Mass: 107,04914) (either Hydroxy-benzylcation or hydroxyl-tropylium cation).

Chemical Formula: C23H23BrN3O4+ Exact Mass: 484,08665

m/z: 484.08665 (100.0%), 486.08460 (97.3%), 487.08795 (24.2%), 485.09000 (16.2%), 485.09000 (8.7%), 488.09131 (1.7%), 486.09335 (1.6%), 488.09131 (1.2%), 485.08368 (1.1%), 487.08163 (1.1%), 486.09335 (1.1%)

Thus the hydroxylation is probably not on the hydrazone ring (contrary to the authentic syntetic derivatives).

             In table 4  : For the hydroxylated molecule : Chemical Formula: C23H23BrN3O5+

Exact Mass: 500,08156 correct value. Thus the orbitrap seems correctly calibrated.

            Figure 6. Your figure is not very interesting since it has little explanation. If Xcalibur was used you could show much better figures. There is a function in Edit Special allowing to copy images from the screen in a precise dimension. Then you could show extracted chromatogram at 484 and 500 values (use exact mas and 10 mmu width). I suggest to Show  the MS and the MS2 spectra. They are interesting to the reader. Your actual plots (TIC) give very little information.

the legend of figure 6 not complete. You should state what is represented.

I suppose that the above panel is the TIC (total ion current). The lower panel is I suppose the trace of the parent drug 11b. Is it MS signal or UV signal. If MS state the value od the ion monitored.

With a Q-exactive Orbitrap you have the standard software Xcalibur that allow showing reconstituted chromatogram at any MS value. It would be of interest to show the reconstituted chromatogram for the parent compound  and all the found metabolites.

For instance You could show m/z = 484.O86 +/_ 10 mmu,

then 500.082 +/- 10 mmu and also the glucuronide and the glycine conjugate.

Then you could show for each compound and metabolite the Mass spectrum and its MS2 spectrum. For the principal metabolites seen,  I would put it in the manuscript. For the other as supplementary material.

            Around line 620-630 : positive ESI 632 - 176 = 556 (corresponds to 584 - 28 thus the carboxylic acid. Probably the acylglucuronide. For glucuronides the negative ESI would have been a good method. For the glycine conjugate of the acid, 456 +75 = 531 -18 = 513

However the mass data on the glucuronide and glycine derivative are strange: acyl-glucuronide Chemical Formula: C27H27BrN3O10+ Exact Mass: 632,08743

The mass error is very big 57 mmu. either the Orbitrap has not been calibrated for weeks or this is not the claimed metabolite.

glycine conjugate Chemical Formula: C23H22BrN4O5+ Exact Mass: 513,07681

Here again there is an error of more than 40 mmu : too much.

         Line 627 : This fragment a MW =  Chemical Formula: C3H7NO2Exact Mass: 89,04768 this is too far off. From the found value. Using Xcalibur one find C4H9O2+ Chemical Formula: C4H9O2 Exact Mass: 89,06025 : that is much closer to the found value.

            Line 338 : dihydroxybenzaldehyde not dy.

Thus I think that the paper is not well constructed.  I have done many critics that could be used by the authors to rewrite their study.  They must discuss with their mass spectroscopist and perhaps do a few more incubations with LC/MS analysis in order to ge more informations on the metabolites. I would suggest doing one incubation and taking probes at 0, 30, 60, 120. And 180 min and analyzing them by HPLC/MS. In principle theuy should get enough data to answer several of my questions. In particulatr they should spend more time in analyzing the raw data of the HPLC/MS experiments.

As a conclusion I think that the paper is not ready for publication and must be rewritten after some complementary studies.

Comments on the Quality of English Language

english is correct. Check the spelling.

Round 2

Reviewer 2 Report

Comments and Suggestions for Authors

I have reviewed my previous comments on this manuscript and I am happy that the authors have addressed them all. I don't think that they quite understood the comments about numbering but at least I can now see what the molecules are. Compound numbers should be in bold throughout. Characterisation data should also be added for 11b. I am confident that the authors can make these changes and I don't need to see the paper again. 

Happy to recommend acceptance, provided, of course, that the authors can also satisfy the requirements of any other reviewers.

Author Response

Dear reviewer, thank you for your interest in our manuscript and for the usefull recommendations. 

As suggested by you all the compounds IDs are checked and replaced with bold font where necessary.

The charactertistics of 11b are included in materials and methods section.

Best regards

Reviewer 3 Report

Comments and Suggestions for Authors

Tehe authors have partially answered to my concerns.

Initially I could to see their supplementary figures showing the MS/MS spectra using Word on my Mac. However I solved it using Open Office.

Strangely they give on this figure the MS2 on the peak of the 81 isotopic peak of bromine.

The 304 peak coresponds to framentind the 2 CO-X bonds to give a conjugated diketene. The 136 peak corresponds to the splitting of the hydrazone to a protonated aromatic nitrile.

When reading carefully the paper and the answer of the authors this is a prelaminary paper showing that the metabolite of their molecule 11b can be analysed by UV HPLC.

It would be clearer if the authors explained better the UV chromatogram of Figure 2 (you still do not describe well the chromatogramm.
If I understand the peak at 13.88 is the parent 11B?
Then you have 3 metabolites at 11.7 min, then 4.66 min and 3.40 min.
The peak at 3.40 min is the hydrazide 11 (refer to the scheme 2) .`
Name the cpmpounds on the figures 11B,  M1, M2, M3 (or any other name) this would be clearer.)

It is astonishing that you did not do a HPLC MS on an actual incubation. You probably would have got informations on the metabolites since your authentic compounds give interesting MS2. This would gide you to synthetisze the potential metabolites.

As a preliminary study, the paper could be accepted after small corrections.

Author Response

Dear reviewer enclosed you will find the proofred recommendations:

It would be clearer if the authors explained better the UV chromatogram of Figure 2 (you still do not describe well the chromatogramm.
If I understand the peak at 13.88 is the parent 11B?
Then you have 3 metabolites at 11.7 min, then 4.66 min and 3.40 min.
The peak at 3.40 min is the hydrazide 11 (refer to the scheme 2) .`
Name the cpmpounds on the figures 11B,  M1, M2, M3 (or any other name) this would be clearer.)

The chromatogram is replaced with a Figure identifying the peaks and the corresponding retention times for each product.

We thank you for your helpfull and thorough review. 

Best regards